# National consensus on entrustable professional activities for competency-based training in anaesthesiology

**Alexander Ganzhorn, Leonie Schulte-Uentrop, Josephine Küllmei, Christian Zöllner, Parisa Moll-Khosrawi*******

Department of Anaesthesiology, University Medical Centre Hamburg-Eppendorf, Hamburg, Germany

\* pmollkho@icloud.com

**Data Availability Statement:** Data is provided in the paper and its supporting information.

**Funding:** The author(s) received no specific funding for this work

## Abstract

Entrustable Professional Activities (EPA) are specialty specific tasks or responsibilities, combining the clinical workplace and the long-demanded competency-based medical education. The first step to transform time-based into EPA-based training is to reach consensus on core EPAs that describe sufficiently the workplace. We aimed to present a nationally validated EPA-based curriculum for postgraduate training in anaesthesiology. Using a predefined and validated list of EPAs, we applied a Delphi consensus approach, involving all German chair directors of anaesthesiology. We then conducted a subsequent qualitative analysis. Thirty-four chair directors participated in the Delphi survey (77% response) and twenty-five completed all the questions (56% overall response). Reflected by the intra-class-correlation, the consensus on the importance (ICC: 0.781, 95% CI [0.671, 0.868]) and the year of entrustment (ICC: 0.973, 95% CI [0.959, 0.984]) of each EPA reached high levels of agreement among the chair directors. The comparison of data assessed in the preceding validation and present study showed excellent and good levels of agreement (ICC entrustment: 0.955, 95% CI [0.902, 0.978]; ICC importance: 0.671, 95% CI [-0.204, 0.888]). The adaptation process, based on the qualitative analysis, resulted in a final set of 34 EPAs. We present an elaborate, fully described and nationally validated EPA-based curriculum, reflecting a broad consensus among different stakeholders of anaesthesiology. We hereby provide a further step towards competency-based postgraduate anaesthesiology training.

## Introduction

Entrustable Professional Activities (EPAs) have gained great popularity over the past years [1] and have been integrated into many undergraduate and postgraduate training programs worldwide [2]. An EPA is a specialty specific task or responsibility, which should be entrusted to the trainee once a sufficient level of performance has been reached [3]. The concept of EPAs is an effective approach to combine the clinical workplace with competency-based medical education (CBME) [4].

Like in many other fields, in anaesthesiology too, the importance of CBME in postgraduate training has been emphasised: The European Society of Anaesthesiology has endorsed the

**Competing interests:** The authors have declared that no competing interests exist.

transformation of anaesthesiology training into an outcome-based approach, moving from time based- towards competency-based training [5]. So far, only few studies have defined EPAs for anaesthesiology training and even fewer have constructed an elaborate training curriculum based on fully described EPAs [6–11]. Although the findings of these studies do contribute to the process of implementing EPA-based training, nevertheless, still some gaps in research emerge: Only few studies aimed to reach institutional [10] or even national consensus [9] on defined EPAs. Some only focused on the first year [6] or undergraduate training [12]. Furthermore, most of the studies included only the titles and other components of a fully described EPA were omitted. Alas, all components of an EPA are necessary to implement an EPA-based training into the clinical workplace [1]. A fully described EPA encompasses seven domains which, amongst others, include the title, specific descriptions and the stage of training at which the task should be entrusted [1].

According to the agenda of Jonker et al., the first step towards CBME in anaesthesiology training is reaching consensus on relevant EPAs [3]. Nevertheless, investigations aiming to reach national or even international consensus are scarce. In a pioneering study, Wisman-Zwarter et al. provided a list of forty-five EPAs which was validated through a national consensus procedure involving Dutch anaesthesiology program directors. Overall, the current state of research deviates from the desired state. There is a gap in research regarding the definition and validation of EPAs for anaesthesiology training in among others, German-speaking countries. No published investigation has reached consensus on EPAs for anaesthesiology training in Germany, much less in Europe [10]. Yet, reaching a broad consensus on EPAs will facilitate their implementation and acceptance [4]. Consensus should be reached on local and institutional levels (those who work directly with the EPAs), but also on more extended levels which include experts and multiple institutions (nationally or even internationally) [3, 4].

We therefore aimed to present an institutionally and nationally validated EPA-based curriculum for postgraduate training in anaesthesiology, which is in accord with the so far published European EPAs.

In a preceding study we developed and institutionally validated an anaesthesiology core curriculum based on thirty-nine EPAs, which includes the year of entrustment and the importance (ranking) of each EPA to the curriculum. Our curriculum showed a 73% accordance with the EPAs provided from the study of Wisman-Zwarter et al. in the Netherlands [8]. To extend the validation and consensus, chair directors nation-wide were targeted, as they are key figures in implementing EPA-based curricula into the workplace.

Using a national consensus procedure, which involved all German chair directors, the EPAs were validated with respect to their importance and the year of entrustment. The curriculum was adjusted accordingly, resulting in a validated core curriculum for competency-based training in anaesthesiology.

## Methods

A detailed project description was sent to the local Ethic Committee of Hamburg which belongs to the General Medical Council of Hamburg (Ethik-Kommission der Ärztekammer Hamburg, Hamburg, Germany). The project was exempted from the need of approval, as the paragraph 9 of the "Law of Healing Professions, Hamburg" (§ 9 des Hamburgischen Kammergesetzes für Heilberufe) and § 15, section 1 of the medical professional conduct (Berufsordnung für Hamburger Ärzte und Ärztinnen) and the article 6 of the "Declaration of Helsinki" did not apply. Therefore, no necessity of deliberation of the project was seen. This study was a study with- but not on humans.

Participation in the study was voluntary and anonymous. Participants declared their consent to participate by participating (this information was provided alongside the email which explained the study goal).

## Study design

This study is reported in accordance with the STROBE and COREQ guidelines [13, 14].

The starting point and basis of our study was a core curriculum for postgraduate anaesthesiology training, composed of thirty-nine EPAs which was developed and validated in a preceding study [8]. We used the combination of a modified Delphi-method and qualitative analysis (expert group, template- and mapping method, direct content analysis) [15, 16], to reach consensus and to adapt the curriculum (EPAs) accordingly. The Delphi-method is an iterative technique with the goal to collect expert opinion to reach group consensus [17, 18]. Qualitative analysis has gained a broad applicability in the field of medical education and especially for developing EPAs [8, 19].

## Study setting

This study was performed at the Department of Anaesthesiology, University Medical Center Hamburg-Eppendorf. The General Medical Council of each federal state sets the prerequisites for postgraduate specialty training in anaesthesiology. Specialist qualification is reached during residency (five years): The residents are assigned to several anaesthesia workplaces, where they function as the primary executing anaesthesiologist. Supervision is provided by specialists. Based on the residents´ level of training and the anaesthesiologic procedure, the amount of supervision varies from a direct form (being directly involved) to a far indirect form (availability by phone) and is not explicitly defined by law. Therefore, the level of supervision and hence teaching depends on the supervisor and on the resident, who can also actively demand for supervision. Supervision and workplace-based teaching differ in every hospital and are based on local circumstances. The numbers of specialty specific procedures, reaching from a "simple" general anaesthesia to more advanced forms (a.e. one-lung ventilation; anaesthesia for abdominal surgery; anaesthesia for neurosurgery) are recorded in a logbook which is provided by the General Medical Council. Each anaesthesiologic procedure has a minimum number which is determined by the General Medical Council and must be conducted by the resident to reach specialty qualification. When hospitals do not have departments of special surgical disciplines, anaesthesia residents rotate to other hospitals to learn and gather the specific procedures (a.e neurosurgery). After five years of residency and after reaching the minimal numbers for each procedure, the program directors can issue the specialist qualification and the resident can take the specialist examination.

Since 2018 the postgraduate education guidelines have been renewed, including competencies but not EPAs. Furthermore, minimal numbers of conducted anaesthesiologic procedures are still integral parts of specialist qualification. The program is put in practice by training site clusters (hospitals) across Germany, where according to federal regulations, each hospital has a program director, who is responsible for the local training. Amongst others, the chairs of the university hospitals (chair directors) ($n = 47$) have the full training authorisation.

## Sample

The chair directors were eligible to participate in the study due to their broad expertise in anaesthesiology training and because of their representative and determining character. The chair directors have completed their specialty training in anaesthesiology, intensive care, pain-, and emergency medicine. All but three program directors ($n = 44$) of anaesthesiology across

**Table 1. Number and characteristics of the study participants.**

| Study participants of the first validation | | | | |
|---|---|---|---|---|
| All participants | Specialists and Supervising attendings | Attendings and 5th year residents | 4th and 3rd year residents | 2nd and 1st year residents |
| n = 80 | n = 23 | n = 20 | n = 18 | n = 19 |
| mean age: 32.88 yr | mean age: 41.2 yr | mean age: 34.2 yr | mean age: 28.9 yr | mean age: 27.2 yr |
| | female n = 10 | female n = 11 | female n = 11 | female n = 9 |
| | male n = 13 | male n = 9 | male n = 7 | male n = 10 |
| Study participants of the second validation | | | | |
| *Delphi procedure* n = 34 chair directors, specialised in anaesthesiology, intensive care, pain-, and emergency medicine | | | | |
| *Participants of the expert group* n = 4 (n = 2 female; n = 1 male; n = 1 second-year resident; n = 3 anaesthesiology specialists; mean age thirty-five yr | | | | |

Abbreviations: yr = year (age)

Germany were invited to participate in the Delphi survey. The chair directors can be found on the homepage of the German Society of Anaesthesiology and Intensive Care Medicine (DGAI). The chair of the Department of Anaesthesiology of Hamburg was not eligible to participate since he was involved as a researcher in the current study. Two further chair directors were not eligible, because they mainly lead the intensive care units. Thirty-four chair directors participated in the Delphi survey (77% response) and twenty-five completed all the questions (56% overall response). Data was analysed only for complete surveys.

The expert group consisted of four researchers and clinicians of different maturity, with profound knowledge in postgraduate training and medical education. Three of them are anaesthesiology specialists (*n* = 2 female and *n* = 1 male), one of them is a second-year resident (female). The mean age was thirty-five years.

Participants of the preceding study were residents, attendings and consultants of the department of anaesthesiology at the University medical Center Hamburg-Eppendorf.

Table 1. summarises the participants of the current and the preceding study.

## Procedure

An overview of the study procedure is shown in Fig 1.

Before the questionnaire for the consensus procedure (Delphi survey) was drafted, the expert group developed an elaborate curriculum based on the thirty-nine EPAs, meeting the requirements of the AMEE (Association for Medical Education in Europe) guide no 99 [4].

Subsequently, the curriculum was piloted within the department for the duration of six months by *n* = 58 residents. The aim was to analyse the practicability of the curriculum in the daily clinical work setting, in order to detect necessary adaptations before the consensus procedure took place.

After the expert group reconfirmed the EPAs, the questionnaire for the Delphi survey was developed, discussed and piloted. In the following step, the chair directors were contacted by e-mail with the purpose of the study and were provided with background information on the concept of EPAs. The e-mail contained a Weblink via which the anonymous participation in the Delphi survey was enabled.

## Outcome measures

As this study was an initial step to meet the prerequisites for a unification of postgraduate training and a transformation from time-based into competency (EPA) based training, the main goal of the study was to reach national consensus on Entrustable Professional Activities

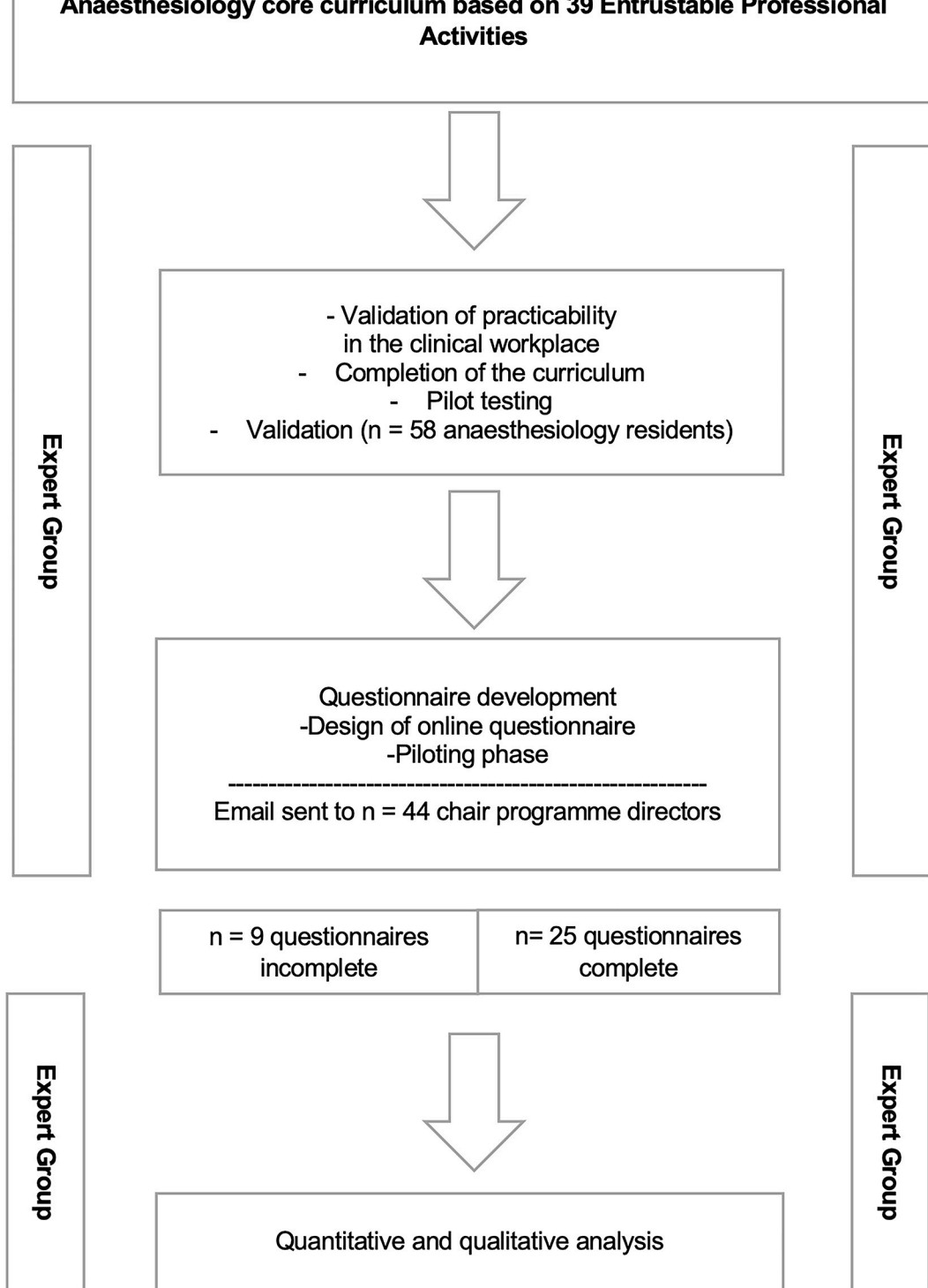

**Fig 1. Study design and procedure.**

for anaesthesiology. Therefore, the first outcome measure was the validation of each EPA, to determine the relevance of each EPA for anaesthesiology training with the primary focus on the content validation. For this purpose, participants were asked to determine, if each EPA should be included in the final list of EPAs. To provide a basis for further adaptations, the importance of each EPA was also numerically assessed to calculate the content validity index (CVI). A CVI reflects the proportion of relevance and is an established parameter in validation processes [20]. The second outcome measure was the year of entrustment for each EPA. Since a broad consensus among different stakeholders of anaesthesiology is helpful to facilitate implementation, the third outcome measure was the consensus between the first two outcome measures (relevance and year of entrustment of each EPA) assessed in the preceding (institutional validation) and current study.

To assess the outcome measures, the Delphi survey was composed of the following three sections (for each EPA):

- Should this EPA be included into an anaesthesiology curriculum? *(Answers: "yes", "no")*

- Please rank each EPA concerning its importance for anaesthesiology training. (Likert- scale-ranks from 5 = *"very important"*, to 1 = *"not important"*).

- In which year of residency should this EPA be entrusted? (conduction without direct supervision) (Answers: "1<sup>st</sup> year, 2<sup>nd</sup> year, 3<sup>rd</sup> year, 4<sup>th</sup> year, 5<sup>th</sup> year, attending")

To provide a basis for adaptations and amendments, the participants were also provided with an open-text box for additional comments and feedback for each EPA.

Data from the Delphi survey was discussed in the expert group regarding necessary amendments of the EPAs. The qualitative data analysis of the open-text comments was conducted, using the mapping method [15], content analysis and an adapted template, which was also used in the preceding study [8, 16]. A triangulation strategy was applied to assess data validity of the comments [21]. First the leader of the research group analysed the comments (preliminary analysis). A second member of the research group repeated independently the same process. Data was then analysed by an independent group of staff members (n = 3). Then the same process was repeated by the expert group and all the amendments were discussed until saturation was reached [22]. For further validation, the results of the expert group were reviewed and confirmed by an independent member of the department.

For the quantitative analysis, agreement of the study participants for each of the three questions (*1. if an EPA should be part of the curriculum; 2. the importance (ranking); 3. the year of entrustment*) was computed. In a next step, the year of entrustment, as well as the relevance (importance) of each EPA (assessed in the present study) was compared to data from the preceding study. Good levels of agreement (intraclass correlation over 0.6) were specified to evaluate the necessity of further changes of the EPA curriculum.

Following thresholds were set and specified (quantitative analysis) for each EPA to pass or to be included into further discussions and considerations:

- 80% agreement among the current study participants on the question, if the EPA should be part of the curriculum was specified to retain an EPA

- Good agreement (reflected by the computed intraclass correlation (ICC) between 0.6 and 0.74) among the study participants regarding the importance of each EPA

- Good agreement among the study participants regarding the year of entrustment

- Good agreement between the study participants of both studies regarding the importance of each EPA

- Good agreement between the study participants of both studies regarding the year of entrustment. If the level of agreement was given, the final year of entrustment was averaged to set the mean value as the final year of entrustment

## Statistical analysis

Statistical analysis was performed with SPSS (version 28.0.1.0, IBM Corp., Armonk, New York, USA). Descriptive statistics (mean values, standard deviations and percentages) were calculated for all data. For the first consensus calculation- which determined if an EPA retained, percentages of agreement ("yes" or "no") were computed and an 80% threshold was specified. The ranking of each EPA was calculated with the content validity index (CVI). The CVI reflects the proportion of relevance [23]. A CVI of 0.75 or higher is considered as "excellent"[24]. According to the CVI, the EPAs were split into a "high ranking"(CVI>0.75) and a "low ranking"group (CVI<0.75) [20].

The consensus (agreement) among the group of chair directors (year of entrustment and ranking) were calculated with the intraclass correlations (ICC): ICC estimates and their 95% confident intervals (95% CI) were calculated based on a mean-rating, absolute-agreement and 2-way mixed-effects model [25].

The levels of agreement with ICC estimates and their 95% CI, between data of both studies (year of entrustment and CVI of each EPA) were calculated based on a test-retest model, mean-rating, absolute-agreement and 2-way mixed-effects model [26].

Values of ICC below 0.40 reflect a poor correlation, between 0.40 and 0.59 a fair correlation, between 0.60 and 0.74 a good correlation and between 0.75 and 1.00 are interpreted as excellent correlation [27].

## Results, ranking and year of entrustment of each EPA

With a mean agreement of 94%, the participants agreed on thirty-seven EPAs to be relevant for anaesthesiology training. Two (5%) EPAs did not reach the 80% consensus rate (EPA: "*Indication and performance of ultrasound use and diagnostic (a.e FAST) and therapeutic consequences*"; EPA: "*Providing perioperative care for patients undergoing cardiothoracic surgery*").

Thirty-five EPAs achieved a CVI over 0.75 (highest CVI: 1.0; lowest CVI: 0.76) and were therefore allocated to the high-ranking group. Four EPAs were allocated to the low-ranking group (highest CVI: 0.72; lowest CVI: 0.6). The two EPAs, which did not pass the 80% consensus rate (to be included in the curriculum) were part of the low-ranking EPAs.

Table 2 displays the content validity index (CVI) of each EPA, dividing them into a high (35/39)- and low/inferior (4/39) ranking group.

The year of entrustment (year of training, in which the EPA can be conducted without direct supervision) is also displayed. For better overview, the CVI and year of entrustment assessed in the preceding study are also outlined.

## Agreement on the CVI and the year of entrustment

The consensus among the chair directors (reflected by the ICC) regarding the importance (ICC: 0.781, 95% CI [0.671, 0.868]) and the year of entrustment (ICC: 0.973, 95% CI [0.959, 0.984]) reached high levels of agreement.

The comparison of data assessed in both studies regarding the year of entrustment and the relevance (CVI) showed excellent and good levels of agreement (ICC entrustment: 0.955, 95% CI [0.902, 0.978]; ICC CVI: 0.671, 95% CI [-0.204, 0.888]).

**Table 2. Content Validity Indices (CVI) and the year (YR) of entrustment of the EPAs.**

| Entrustable Professional Activity | National validation | | Results by Moll-Khosrawi et. al. | |
|---|---|---|---|---|
| | CVI | YR ± SD | CVI | YR ± SD |
| Administer general anaesthesia including regular airway management | 0,96 | 1,16 ± 0,47 | 0,93 | 1,36 ± 0,62 |
| Providing anaesthetic care for small laparoscopic surgery | 0,84 | 1,36 ± 0,57 | 0,88 | 1,66 ± 0,77 |
| Providing perioperative care for patients with ASA ≤ III | 0,96 | 1,40 ± 0,87 | 0,88 | 1,50 ± 0,74 |
| Performing a premedication round (preoperative evaluation) including patient education | 1,00 | 1,48 ± 0,87 | 0,75 | 1,68 ± 1,00 |
| Indication and administration of blood transfusion | 0,96 | 1,68 ± 0,85 | 0,91 | 2,03 ± 0,91 |
| Indication and performance of a regional anaesthesia technique | 0,96 | 2,12 ± 0,73 | 0,68 | 2,89 ± 1,11 |
| Providing postoperative pain management | 1,00 | 2,24 ± 0,97 | 0,76 | 2,20 ± 1,16 |
| Indication, consideration and performance of spinal and epidural anaesthesia | 0,96 | 2,32 ± 0,80 | 0,8 | 2,43 ± 0,88 |
| Providing postoperative care in the recovery room | 0,92 | 2,48 ± 1,12 | 0,85 | 2,09 ± 0,94 |
| Indication and performance of analgosedation | 0,8 | 2,56 ± 0,92 | 0,73 | 2,16 ± 0,89 |
| Providing anaesthetic care for intracranial surgery without increased intracranial pressure | 0,92 | 2,92 ± 0,91 | 0,8 | 2,60 ± 1,01 |
| Providing anaesthetic care for large laparoscopic surgery | 0,72 | 3,00 ± 0,91 | 0,85 | 2,58 ± 1,09 |
| Administer general anaesthesia in patients with increased risk of aspiration | 0,92 | 3,04 ± 1,27 | 0,96 | 3,08 ± 1,59 |
| Providing anaesthetic care for extensive open abdominal surgery | 0,96 | 3,24 ± 0,72 | 0,94 | 3,38 ± 0,90 |
| Performing in-house transfers of critically ill patients | 0,84 | 3,36 ± 0,99 | 0,79 | 2,63 ± 0,95 |
| *Indication and performance of ultrasound use and diagnostic (a.e FAST) and therapeutic consequences | 0,6 | 3,56 ± 1,19 | 0,44 | 3,63 ± 1,26 |
| Providing perioperative care for patients with ASA > III | 1,00 | 3,64 ± 0,91 | 0,89 | 2,74 ± 1,05 |
| Administer general anaesthesia in pediatric patients over the age of five | 0,88 | 3,64 ± 0,91 | 0,8 | 3,60 ± 1,07 |
| Providing anaesthetic management for pregnant patients | 0,96 | 3,64 ± 1,04 | 0,75 | 3,24 ± 1,02 |
| Administer general anaesthesia including airway management in patients with anticipated difficult airway | 1,00 | 3,72 ± 1,24 | 0,94 | 3,41 ± 1,36 |
| Providing perioperative coagulation management including interpretation and therapeutical consequences of thrombelastometry | 0,84 | 3,76 ± 1,01 | 0,61 | 3,88 ± 1,26 |
| Haemodynamic management of major blood loss | 1,00 | 3,76 ± 1,13 | 0,94 | 3,38 ± 1,15 |
| Providing emergency care for critically injured and ill patients during in-house transfers and diagnostic procedures | 0,8 | 3,80 ± 0,87 | 0,76 | 3,86 ± 0,95 |
| Management of the unanticipated difficult airway | 1,00 | 3,84 ± 1,40 | 0,89 | 3,73 ± 1,64 |
| Providing anaesthetic care for thoracic surgery (including lung seperation) with normal lung function | 0,92 | 3,88 ± 0,78 | 0,81 | 3,30 ± 0,91 |
| Communication with relatives of critically ill patients and consultation about treatment plans | 0,88 | 3,88 ± 0,83 | 0,70 | 2,45 ± 1,28 |
| Indication of anaesthetic technique and performance of medullary or general anaesthesia for regular and emergency caesarean section | 1,00 | 4,00 ± 1,08 | 0,89 | 3,51 ± 1,07 |
| Management of in-house emergencies | 1,00 | 4,08 ± 0,81 | 0,79 | 3,93 ± 1,02 |
| Providing anaesthetic care for intracranial surgery with (the risk of) increased intracranial pressure | 0,88 | 4,08 ± 1,00 | 0,71 | 3,66 ± 1,06 |
| Providing anaesthetic care for patients with severe pre-existing cardiac conditions | 0,96 | 4,32 ± 0,69 | 0,78 | 3,84 ± 1,06 |
| Administer general anaesthesia in pediatric patients under the age of five | 0,84 | 4,40 ± 0,76 | 0,64 | 4,43 ± 0,98 |
| Providing anaesthetic and haemodynamic management of uterine atony | 0,96 | 4,44 ± 0,92 | 0,74 | 4,61 ± 1,17 |
| Providing anaesthetic care and emergency management for critically injured and ill patients in the shock room | 0,92 | 4,60 ± 0,76 | 0,78 | 4,34 ± 1,00 |
| Providing anaesthetic care for pregnant patients with HELLP/pre-eclampsia/eclampsia | 0,84 | 4,60 ± 0,76 | 0,58 | 4,40 ± 1,21 |
| Providing perioperative care for critically injured patients with increased intracranial pressure | 0,92 | 4,76 ± 0,44 | 0,69 | 4,41 ± 1,09 |
| Providing anaesthetic care for thoracic surgery (including lung seperation) with limited lung function | 0,76 | 4,76 ± 0,44 | 0,63 | 4,27 ± 1,09 |
| Administer general anaesthesia in neonatal patients | 0,68 | 4,80 ± 0,50 | 0,45 | 5,28 ± 0,94 |
| Providing perioperative care for patients with major blood loss and pre-existing coagulation disorder | 0,96 | 4,88 ± 0,33 | 0,71 | 4,39 ± 1,20 |
| *Providing perioperative care for patients undergoing cardiothoracic surgery | 0,48 | 4,88 ± 0,33 | 0,28 | 4,70 ± 0,89 |

*Note*: The EPAs are displayed in the chronological order of entrustment; *EPAs that did not reach the 80% consensus rate and are not included in the final curriculum.

*Abbreviations*: CVI = content validity index; YR = year in which an EPA should be conducted by the trainee without direct supervision; SD = standard deviation;

ASA = American Society of Anaesthesiologists; FAST = focused assessment with sonography for trauma; HELLP = hemolysis, elevated liver enzymes, low platelet count

As displayed in Table 2. the chair directors ranked the EPAs of higher relevance (CVI) than in the preceding study with one exception: The EPA *"Providing anaesthetic care for large laparoscopic surgery"* achieved an CVI of 0.72 (low-ranking) whereas in the preceding study a CVI of 0.85 (high-ranking).

### Qualitative analysis and alignment process

A total of *n* = 137 comments were provided by the study participants. All comments were analysed and *n* = 42 comments were considered for the adaptation and adjustment process. A summarised description of the alignment process is provided in the appendix (S1 Appendix).

The alignment process led to four major changes of the curriculum: Two EPAs *("Indication and performance of ultrasound use and diagnostic (a.e FAST) and therapeutic consequences"; "Providing perioperative care for patients undergoing cardiothoracic surgery")* were excluded from the core curriculum. First because they did not pass the 80% mark and secondly because the study participants declared these EPAs to be too specific for general training in anaesthesiology. The two EPAs *"Communication with relatives of critically ill patients and consultation about treatment plans"; "Performing a premedication round (preoperative evaluation) including patient education"* were merged to *"Performing a premedication round (preoperative evaluation) including patient education and communication with relatives"*, as the comments showed that there are many intersections between the competencies of those EPAs. The EPA *"Providing emergency care for critically injured and ill patients during in-house transfers and diagnostic procedures"* received comments that categorized its competencies into the EPA *"Performing in-house transfers of critically ill patients"*. Therefore, the two EPAs were merged and the competencies nested into one extended EPA *("Performing in-house transfers of critically ill **and injured** patients")*. The EPA *"Providing perioperative care for critically injured patients with increased intracranial pressure"* is composed of competencies which are already covered in the EPAs *"Providing anaesthetic care for intracranial surgery with (the risk of) increased intracranial pressure"* and *"Providing anaesthetic care and emergency management for critically injured and ill patients in the shock room"*. Therefore, the EPA *"Providing perioperative care for critically injured patients with increased intracranial pressure"* was nested in the two above mentioned and in the EPA which describes the management of critically injured and ill patients. The EPA was extended with associated competencies in the operating room, resulting in an amended EPA: *"Providing anaesthetic care and emergency management for critically injured and ill patients in the shock room and OR"*.

The adaptation and adjustment process resulted in a curriculum consisting of *n* = 34 EPAs, which is depicted in Fig 2.

## Discussion

In our prospective multi-step national consensus study, 25 experts of anaesthesiology across Germany reached a high degree of agreement on an EPA-based curriculum, which includes the year of entrustment as well as the ranking for each EPA (reflecting its importance). The curriculum was generated and pre-validated in a preceding study and initially included 39 EPAs. After the national consensus procedure (Delphi-method) and a comprehensive qualitative analysis, the curriculum was re-validated. The modifications and amendments resulted in a curriculum composed of 34 EPAs, which constitute a comprehensive range of relevant perioperative- and emergency anaesthetic care.

A strength of our study is the broad consensus on the curriculum, which has been reached from a diverse group of different stakeholders including learners as well as educators. The first validation of the preceding study included clinicians in different stages of training, who are

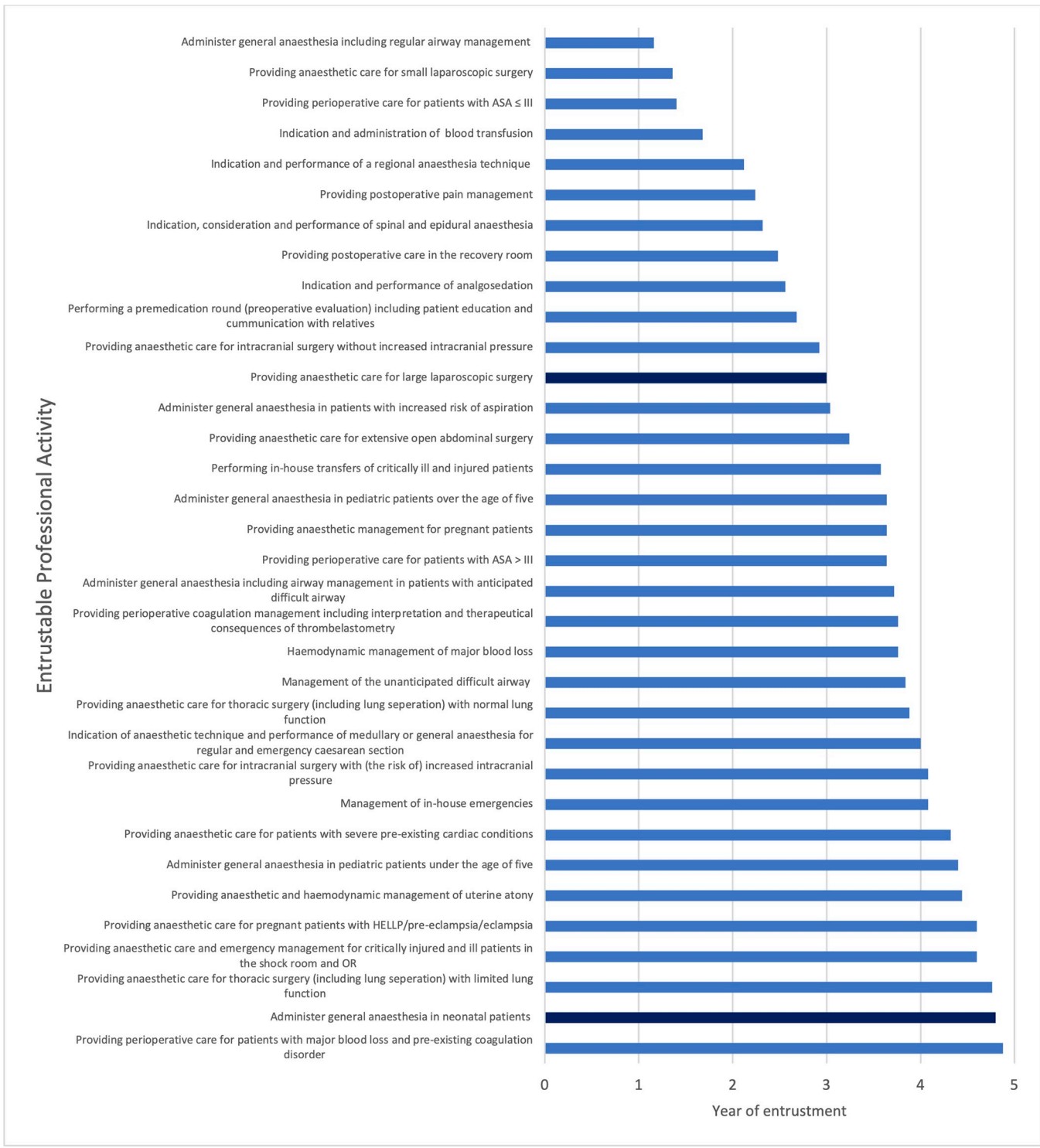

**Fig 2. Entrustable professional activites displayed by the year of entrustment.** Note: The year of entrustment is indicated for each EPA; the light-blue bars display the high-ranking EPAs; the dark-blue bars display the low-ranking EPAs. Abbreviations: ASA = American Society of Anaesthesiologists; HELLP = hemolysis, elevated liver enzymes, low platelet count.

mostly concerned by training curricula on a daily work basis (a.e attendings- functioning as educators and supervisors and trainees as the actual learners) [8]. For the validation of the present study, we have gone a step further and included the German anaesthesiology chair directors. Taking into account that the results of the EPA ranking and year of entrustment of both studies reached high levels of agreement, an extended consensus can be assumed. This broad consensus is important, because an EPA-based curriculum should create a shared mental model amongst learners and educators, resulting in greater acceptance. This would in turn facilitate its implementation and also support learner-oriented teaching [1, 10, 28]. Furthermore, the inclusion of a wide range of study participants prevented a limited and partial view of the subject (curriculum) and hereby also differentiates our study from recently published works on EPAs, which either included program directors or members of a department (trainees and supervisors) [6, 8, 10].

Interestingly, with one exception, the chair directors rated the EPAs to be more important, than they were rated in the preceding study, which resulted in higher CVIs (reflecting the importance of each EPA). Whilst remaining conscious that the levels of agreement between the CVIs of both studies were still good (ICC: 0.671), the discrepancy of the CVIs can be explained in two respects: On the one hand, different experience levels might have caused different perspectives on the EPAs, leading to higher estimates of importance (CVI), assessed within the group of greater experienced (chair directors). On the other hand, the lower estimates assessed in the preceding study might be based on the Dunning-Kruger effect [29]. A cognitive bias whereby low experienced individuals prone to overestimate their abilities. Therefore, the lower experienced anaesthesiologists from the preceding study, might have not overlooked the importance and hidden hazards of a clinical tasks, resulting in lower CVIs [29]. This theory can be supported exemplary with the CVI differences for the EPAs: *"Performing a premedication round including patient education"* or *"Postoperative pain management"*. While a trainee might not realise the relevance and importance, the expert values the outcome of good competencies while conducting the task.

Some of the discrepancies between the CVIs might be due to differing weighting and importance of the competencies at the departments. This argument can be also transferred to the EPA *"Providing anaesthetic care for large laparoscopic surgery"*, that received a higher CVI in the preceding study. The CVI was assessed at a department where laparoscopic procedures are of great significance. Nevertheless, all included EPAs of the new curriculum passed the more important question, if they should be integrated into the curriculum and therefore, the value of the CVI lies within the purpose of further adaptations.

One limitation of our study is that the initial curriculum was developed at a university medical center (maximum care hospital), which might have biased some perceived daily tasks. To circumvent this potential bias and to refine the curriculum accordingly, the study participants were asked to provide feedback for each EPA in an open text field. These comments led to adaptations of the curriculum, making it more transferable to non-maximum care hospitals. Furthermore, the amendments might reduce difficulties in using our results for a curriculum reform of other faculties, as we moved from an initial single center design towards a broader and more representative validation [30].

The amendments resulted in a reduction of the curriculum to thirty-four EPAs. Two EPAs were excluded from the core curriculum, as they did not reflect relevant daily tasks of the anaesthesiologic workplace (EPAs: *"Indication and performance of ultrasound use and diagnostic (a.e FAST) and therapeutic consequences"; "Providing perioperative care for patients undergoing cardiothoracic surgery"*). Furthermore, many comments indicated that too many EPAs of the curriculum target competencies of the care of critically ill and injured patients. Therefore, some of these EPAs were merged and their competencies nested into other EPAs. This

quantitative reduction of the EPAs purses the recommendation which describes an amount of twenty to thirty EPAs per curriculum [31]. Considering the extent of our curriculum, one might argue that the number of EPAs constitute a further limitation of our study. Nevertheless, taking into account the large range of so far published EPA curricula, which include 10 to 45 EPAs [32], as well as the fact that anaesthesiology training is a comprehensive program with a lot of competencies, the number of our EPAs can be justified.

A further strength of our study is that we truly met the challenges of the concept of EPAs. We took into account and transposed the specific recommendations, that were provided by ten Cate and his co-workers regarding the development of an EPA-based curriculum [1, 4]. Considering these recommendations is an important aspect, which has been neglected in some works. As recently described, there has been a diversity in the use of the term of EPAs, diluting the concept as intended and hereby hampering the values of EPA in CBME [33].

Although the content of our curriculum itself is not generalizable to medical education, the value of the approach of our study still merits consideration. This approach could assist other medical disciplines, in graduate and undergraduate medical education, with the integration of EPAs into training.

## Conclusion

Interest in EPAs is rapidly growing and their potential to facilitate the implementation of CBME into the clinical workplace has been emphasised [34, 35]. Nevertheless, few are used systematically in anaesthesiology training [36], mainly due to little availability and also because this novel approach is not well understood [37]. With our study a further step is made towards the transformation of anaesthesiology training into a more contemporary approach, by presenting a validated set of EPAs and by introducing this concept to many stakeholders of anaesthesiology. Our curriculum does not only provide specific EPA titles, but includes the importance of each EPA, the year of entrustment as well as other important components. Thereof, our study presents a broadly validated EPA-based curriculum for German speaking countries, which has also reached a 73% agreement with validated EPAs published in the Netherlands [10]. Hereby, our results provide a further base towards unifying the goal of CBME in anaesthesiology.

## Supporting information

**S1 Appendix. Alignment process of the EPAs.**
(PDF)

## Acknowledgments

We would like to thank the German chair directors for participating in this study. We would like to thank all members of the Department of Anaesthesiology at the University Medical Center Hamburg-Eppendorf for their support.

## Author Contributions

**Conceptualization:** Alexander Ganzhorn, Christian Zöllner, Parisa Moll-Khosrawi.

**Data curation:** Alexander Ganzhorn, Josephine Küllmei, Parisa Moll-Khosrawi.

**Formal analysis:** Alexander Ganzhorn, Leonie Schulte-Uentrop, Josephine Küllmei, Parisa Moll-Khosrawi.

**Investigation:** Alexander Ganzhorn, Leonie Schulte-Uentrop, Josephine Küllmei, Christian Zöllner, Parisa Moll-Khosrawi.

**Methodology:** Leonie Schulte-Uentrop, Christian Zöllner, Parisa Moll-Khosrawi.

**Project administration:** Parisa Moll-Khosrawi.

**Resources:** Parisa Moll-Khosrawi.

**Supervision:** Leonie Schulte-Uentrop, Christian Zöllner, Parisa Moll-Khosrawi.

**Validation:** Alexander Ganzhorn, Josephine Küllmei, Parisa Moll-Khosrawi.

**Visualization:** Alexander Ganzhorn, Parisa Moll-Khosrawi.

**Writing – original draft:** Alexander Ganzhorn, Parisa Moll-Khosrawi.

**Writing – review & editing:** Leonie Schulte-Uentrop, Josephine Küllmei, Christian Zöllner, Parisa Moll-Khosrawi.

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
