## [Editor Report · Decision Letter 0]

2 Dec 2022

PONE-D-22-30955National consensus on Entrustable Professional Activities for competency-based training in AnaesthesiologyPLOS ONE

Dear Dr. Moll-Khosrawi,

Thank you for submitting your manuscript to PLOS ONE. After careful consideration, we feel that it has merit but does not fully meet PLOS ONE’s publication criteria as it currently stands. Therefore, we invite you to submit a revised version of the manuscript that addresses the points raised during the review process.

We look forward to receiving your revised manuscript.

Kind regards,

Shazia Iqbal

Academic Editor

PLOS ONE

Journal Requirements:

3. You indicated that ethical approval was not necessary for your study. We understand that the framework for ethical oversight requirements for studies of this type may differ depending on the setting and we would appreciate some further clarification regarding your research. Could you please provide further details on why your study is exempt from the need for approval and confirmation from your institutional review board or research ethics committee (e.g., in the form of a letter or email correspondence) that ethics review was not necessary for this study? Please include a copy of the correspondence as an ""Other"" file.

Additional Editor Comments:

Title; National consensus on Entrustable Professional Activities for competency-based

training in Anaesthesiology is relevant and Effective and represents the Main Message of the Paper

Introduction

• In the introduction section, the background of the research topic is mentioned appropriately as you have mentioned that few studies have defined EPAs for anaesthesiology training and even fewer have constructed an elaborate training curriculum based on fully described EPAs. However, it is not explicit what are research gaps. Highlight possibilities out of these few studies to strengthen the problem statement. Also, mention a clear problem statement that emphasizes the reason for you to study this research topic.

• The rationale of the study is explicit and justified. However, it is better to elaborate more that why you’re planning to choose the national consensus procedure rather than other options like institutional consensus.

• The objective of the study is clear, concise, and complete.

• The professional significance of the study is clearly given

Material and Methods

• Setting and Duration are described but insufficient information regarding the workplace, so elaborate more.

• Inclusion and Exclusion criteria are specified, and a description of participants is mentioned. However, the Characteristics of the participants (25 experts in anaesthesiology) need to be summarized in table form.

• Investigation/Procedure/Technique is clearly given. The study design is clear, however, the Important Variable Measurement Criteria need to be specified-outcome. Elaborate more on the Criteria to measure outcome relevance.

Results

• Statistical analysis is mentioned and the Presentation of results in table 1 is clear (Table 1. Content Validity Indices (CVI) and the year (YR) of entrustment of the EPAs), however, the basic demographic data of participants in the study (age, gender, education, background, work experience level, etc) is not mentioned. It’s important to mention the first table before the study. Sometimes, this data is linked to the outcome of study measures.

As you have mentioned in the discussion ( On the one hand, different experience levels might have caused different perspectives on the EPAs, leading to higher estimates of importance (CVI), assessed within the group of greater experienced (chair directors). On the other hand, the lower estimates assessed in the preceding study might be based on the Dunning-Kruger effect (27). A cognitive bias whereby low-experienced individuals are prone to overestimate their abilities. Therefore, the lower experienced anesthetists from the preceding study might have not overlooked the importance and hidden hazards of clinical tasks, resulting in lower CVIs (27).)

• Data satisfies the objectives of the study, all subjects/Variables in the study are accounted for appropriate statistical tests are applied, and Data Analysis is correctly interpreted. The alignment process led to four major changes in the curriculum is explicit. However, Figures 1 and 2 are not illegible in this pdf so cannot comment. Make it in readable format

Discussion

Overall, Important findings and results were discussed and New emerging findings discussed

• Results of the study discussed with previous literature

• Results and analysis leads to firm conclusions

• Limitations of the study mentioned, however, there are no comments on the generalization of the results in medical education.

Conclusions

• Conclusion justified and based on the critical argument

References

• Written in the required style however, Mostly are more than 5 years old so use the latest research studies, especially in the discussion part, and add Local literature.

• Few Grammatical mistakes need careful review.

---

## [Author Response · Author response to Decision Letter 0]

14 Feb 2023

Please find our point-by-point responses below. The page and line details refer to the document which includes the track changes.

Note: Regarding the data availability statement, we would like to state that there is no further repository in-formation. All information has been included within the manuscript and the supplementary material. If despite our provided summary of comments access to all comments is desired, we will certainly provide them.

Sincerely,

Parisa Moll-Khosrawi (on behalf of all authors)

Editorial comments and responses and journal requirements

Journal requirements

Please provide additional details regarding participant consent. In the ethics statement in the Methods and online submission information, please ensure that you have specified what type you obtained (for instance, written or verbal, and if verbal, how it was documented and witnessed). If your study included minors, state whether you obtained consent from parents or guardians. If the need for consent was waived by the ethics committee, please include this information.

Response: As participation in the study was voluntary and anonymous, no personified informed consent was obtained. Participants were informed that by filling out the questionnaire (participation), they declared their consent to participate within the study. This statement also included the publication of data generated within the study. For more clarification, we have extended the methods section with an explanation. Please find the paragraph on page 5, lines 119-121.

“Participation in the study was voluntary and anonymous. Participants declared their consent to participate by participating (this information was provided alongside the email which explained the study goal).” 

You indicated that ethical approval was not necessary for your study. We understand that the framework for ethical oversight requirements for studies of this type may differ depending on the setting and we would ap-preciate some further clarification regarding your research. Could you please provide further details on why your study is exempt from the need for approval and confirmation from your institutional review board or re-search ethics committee (e.g., in the form of a letter or email correspondence) that ethics review was not nec-essary for this study? Please include a copy of the correspondence as an ""Other"" file.

Response: We contacted the local Ethic Committee of Hamburg with a detailed project description. The project exempted from the need of approval, as the paragraph 9 of the "Law of Healing Professions, Hamburg” (§ 9 des Hamburgischen Kammergesetzes für Heilberufe), § 15 section 1 of the medical professional conduct (Berufsordnung für Hamburger Ärzte und Ärztinnen) and the article 6 of the “Declaration of Helsinki” did not apply. Therefore, no necessity of deliberation of the project was seen (this study was a study with- but not on humans). We have attached the email correspondence (“other file”) with the Ethic Committee, which exempts the project from the need of approval. For more clarity we have amended the paragraph on the ethical approv-al (pages 4 lines 112-118).

“A detailed project description was sent to the local Ethic Committee of Hamburg which belongs to the General Medical Council of Hamburg (Ethik‐Kommission der Ärztekammer Hamburg, Hamburg, Germany). The project was exempted from the need of approval, as the paragraph 9 of the "Law of Healing Professions, Hamburg” (§ 9 des Hamburgischen Kammergesetzes für Heilberufe) and § 15, section 1 of the medical professional conduct (Berufsordnung für Hamburger Ärzte und Ärztinnen) and the article 6 of the “Declara-tion of Helsinki” did not apply. Therefore, no necessity of deliberation of the project was seen. This study was a study with- but not on humans.” 

Editor comments and responses

Title; National consensus on Entrustable Professional Activities for competency-based

training in Anaesthesiology is relevant and Effective and represents the Main Message of the Paper

Introduction

Response: Thank you- we are happy to meet your satisfaction.

In the introduction section, the background of the research topic is mentioned appropriately as you have men-tioned that few studies have defined EPAs for anaesthesiology training and even fewer have constructed an elaborate training curriculum based on fully described EPAs. However, it is not explicit what are research gaps. Highlight possibilities out of these few studies to strengthen the problem statement. Also, mention a clear problem statement that emphasizes the reason for you to study this research topic.

Response: Thank you very much for this helpful comment. We have now worked out clearly and presented in more detail the current state and findings of EPA research (specifically for anaesthesiology). Based on that we identified the research gaps and hereby strengthen our problem statement. According to our research, there simply is no elaborated study or work which targeted to develop an EPA-based curriculum for postgraduate (even undergraduate) training in anaesthesiology. The existing studies often include solely the EPA titles. Alas, a fully described EPA includes more than just the title. To facilitate implementation and acceptance of EPA-based training into the clinical workplace, fully described EPAs are needed (there is an elaborate description by Ten Cate in the AMEE Guide n. 99, which we have cited). Besides of a description that matches the criteria men-tioned by Ten Cate et al., the EPAs need to be accepted by many stakeholders- so a validation is needed. There-fore, our statement (there is no elaborate curriculum based on fully described EPAs) can be considered from two angles: First, as we have stated, no published study included all components of an EPA (seven in total) that are necessary for an EPA to be complete. Most of the studies defined EPA titles (one component) and omitted the other components. In order to transform postgraduate training into an EPA-based approach all components of an EPA are necessary. Secondly, only few EPAs are truly validated. Wisman-Zwarter et al. validated the EPAs nationally- but the EPAs do not include all components. Marty et al. validated EPAs for the first year institution-ally- also not including all components. For the sake of completeness, we would like to mention the work of the US colleagues Woodworth et al., who have developed a digitalized EPA based curriculum, which includes titles and procedural skills but omits theoretical skills. Nevertheless, there is still a difference to our curriculum- we have included more skill domains of an EPA and Woodworth et al. did not validate the EPAs nationally, including many stakeholders. Furthermore, due to differences in postgraduate training in the US and EU, the curriculum cannot be transferred to European countries. Taking all aspects into account, the need for our work is clear and the gap in research is evident. As the current state of EPA-based training in anaesthesiology differs from the desired state, we took a step towards unifying the goal of CMBE in anaesthesiology postgraduate training and hereby use the concept of EPAs as it has been intended (including all EPA components and validating the curric-ulum broadly).

Please find the amended paragraph on page 3, lines 64-73: 

“So far, only few studies have defined EPAs for anaesthesiology training and even fewer have constructed an elaborate training curriculum based on fully described EPAs (6-11). Although the findings of these studies do contribute to the process of implementing EPA-based training, nevertheless, still some gaps in research emerge: Only few studies aimed to reach institutional (10) or even na-tional consensus (9) on defined EPAs. Some only focused on the first year (6) or undergraduate training (12). Furthermore, most of the studies included only the titles and other components of a fully described EPA were omitted. Alas, all components of an EPA are necessary to implement an EPA-based training into the clinical workplace (1). A fully described EPA encompasses seven domains which, amongst others, include the title, specific descriptions and the stage of training at which the task should be entrusted (1).”

Furthermore, the problem statement emerges, as CBME has been demanded by several position papers, amongst others, by the European Society of Anaesthesiology (ESA). In an agenda of the ESA, the first step to-wards CBME in anaesthesiology has been stated to be consensus on relevant EPAs. Taking together, that no elaborate EPA-based curriculum exists which has been validated (neither on an institutional- nor on a national level), underlines the need for our research. 

Please find a new paragraph which includes the problem statement, highlighting the reason for us to study this research topic:

“According to the agenda of Jonker et al., the first step towards CBME in anaesthesiology training is reaching consensus on relevant EPAs (3). Nevertheless, investigations aiming to reach national or even international consensus are scarce. In a pioneering study, Wisman-Zwarter et al. provided a list of forty-five EPAs which was validated through a national consensus procedure involving Dutch anaesthesiology program directors. Overall, the current state of research deviates from the desired state. There is a gap in research regarding the definition and validation of EPAs for anaesthesiology training in among others, German-speaking countries. No pub-lished investigation has reached consensus on EPAs for anaesthesiology training in Germany, much less in Europe (10). Yet, reaching a broad consensus on EPAs will facilitate their implementation and acceptance (4). Consensus should be reached on local and insti-tutional levels (those who work directly with the EPAs), but also on more extended levels which include experts and multiple institu-tions (nationally or even internationally) (3, 4)”

(Pages 3,4; lines 76-87)

The rationale of the study is explicit and justified. However, it is better to elaborate more that why you’re plan-ning to choose the national consensus procedure rather than other options like institutional consensus.

Response: We apologize for the lack of clarity that we have caused due to our wording. Our curriculum has already been validated institutionally in a preceding study. In order to facilitate implementation and acceptance of an EPA-based curriculum, a broad consensus should be reached, involving different stakeholders of (in this case) anaesthesiology. Therefore, an institutional, as well as a national consensus is important. The national consensus is the main catalysator in the process of unifying training nationwide and comprehensively. We aimed to reach a broad consensus for our curriculum: Therefore, in the first step (which is referred to as the preceding study)- we reached institutional consensus and in a next step, we conducted the national validation. The national validation which includes key figures (a.e chair directors) is important for the facilitation of future implementation processes. In a further step, all data collected, we investigated the level of agreement (consen-sus) between data from the institutional and national consensus. As there were high levels of agreement be-tween the data we can assume a broad consensus, involving multiple stakeholders. Furthermore, our curriculum shows over 73% agreement with the work of Wisman-Zwarter et al.- who have published EPAs based on na-tional consensus in the Netherlands (even if their EPAs do not include all components, their work is a reasonable approach). 

For more clarification, we have now amended the paragraph in the introduction. Please find it on page 4, lines 90-97: 

“We therefore aimed to present an institutionally and nationally validated EPA-based curriculum for postgraduate training in anaes-thesiology, which is in accord with the so far published European EPAs. 

In a preceding study we developed and institutionally validated an anaesthesiology core curriculum based on thirty-nine EPAs, which includes the year of entrustment and the importance (ranking) of each EPA to the curriculum. Our curriculum showed a 73% accord-ance with the EPAs provided from the study of Wisman-Zwarter et al. in the Netherlands (8). To extend the validation and consensus, chair directors nation-wide were targeted, as they are key figures in implementing EPA-based curricula into the workplace.”

The objective of the study is clear, concise, and complete.

The professional significance of the study is clearly given.

Response: Thank you.

Setting and Duration are described but insufficient information regarding the workplace, so elaborate more.

Response: Thank you for this comment which led to an elaborated description of the workplace and highlights even more the need for unification of training. 

We now describe how specialty qualification is reached by working 5 years as an anaesthesiologist, being as-signed to different workplaces. We highlight how supervision is conducted by specialists and how variable this supervision is. Furthermore, we describe that still- even after the amendments of the postgraduate guidelines (provided by the General Medical Council) - the minimal numbers of conducted procedures are incremental and the concept of CBME is actually neglected. Please find the new paragraph in the method section, on pages 6 and 7, lines 135-157:

“The General Medical Council of each federal state sets the prerequisites for postgraduate specialty training in anaesthesiology. Specialist qualification is reached during residency (five years): The residents are assigned to several anaesthesia workplaces, where they function as the primary executing anaesthesiologist. Supervision is provided by specialists. Based on the residents´ level of training and the anaesthesiologic procedure, the amount of supervision varies from a direct form (being directly involved) to a far indirect form (availability by phone) and is not explicitly defined by law. Therefore, the level of supervision and hence teaching de-pends on the supervisor and on the resident, who can also actively demand for supervision. Supervision and workplace-based teach-ing differ in every hospital and are based on local circumstances. The numbers of specialty specific procedures, reaching from a “sim-ple” general anaesthesia to more advanced forms (a.e. one-lung ventilation; anaesthesia for abdominal surgery; anaesthesia for neurosurgery) are recorded in a logbook which is provided by the General Medical Council. Each anaesthesiologic procedure has a minimum number which is determined by the General Medical Council and must be conducted by the resident to reach specialty qualification. When hospitals do not have departments of special surgical disciplines, anaesthesia residents rotate to other hospitals to learn and gather the specific procedures (a.e neurosurgery). After five years of residency and after reaching the minimal numbers for each procedure, the program directors can issue the specialist qualification and the resident can take the specialist examination. Since 2018 the postgraduate education guidelines have been renewed, including competencies but not EPAs. Furthermore, minimal numbers of conducted anaesthesiologic procedures are still integral parts of specialist qualification. The program is put in practice by training site clusters (hospitals) across Germany, where according to federal regulations, each hospital has a program director, who is responsible for the local training.”

Inclusion and Exclusion criteria are specified, and a description of participants is mentioned. However, the Characteristics of the participants (25 experts in anaesthesiology) need to be summarized in table form.

Response: Thank you for this important and valuable request. As we conducted the study anonymously, no actual demographic data (regarding sex, age, ethnic background etc) of chairs has been collected. Nonetheless, the most important information in the context of our study is provided: All participants are specialized in anaes-thesiology, pain- and emergency medicine, as well as in intensive care. Furthermore, in their position, they are the key figures in determining framework conditions of postgraduate training. As we wanted to follow your reasonable request, we drafted a new table (table 1, page 8) which includes demographic data of study partici-pants of both studies. We actually believe that this table has improved our manuscript, as it is very useful in supporting the overview of the study design for your valuable readers.

Investigation/Procedure/Technique is clearly given. The study design is clear, however, the Important Variable Measurement Criteria need to be specified-outcome. Elaborate more on the Criteria to measure outcome rele-vance.

Response: Thank you very much for this helpful comment. For more clarification, we have added a new para-graph in the methods section, with the subheading “Outcome measures”. Here, we elaborate more on the outcomes, what they are and why we targeted specifically these outcome measures. We explain which kind of validation we aimed to assess- namely content validation. Furthermore, we describe that the year of entrust-ment was assessed and state the reason for this outcome. Finally, we elaborate on the third outcome measure, the agreement between the data assessed from the chairs and between the data from the chairs and the pre-ceding study:

Please find the new paragraph on page 9, lines 194-207:

“As this study was an initial step to meet the prerequisites for a unification of postgraduate training and a transformation from time-based into competency (EPA) based training, the main goal of the study was to reach national consensus on Entrustable Professional Activities for anaesthesiology. Therefore, the first outcome measure was the validation of each EPA, to determine the relevance of each EPA for anaesthesiology training with the primary focus on the content validation. For this purpose, participants were asked to determine, if each EPA should be included in the final list of EPAs. To provide a basis for further adaptations, the importance of each EPA was also numerically assessed to calculate the content validity index (CVI). A CVI reflects the proportion of relevance and is an established parameter in validation processes. (20) The second outcome measure was the year of entrustment for each EPA. Since a broad consensus among different stakeholders of anaesthesiology is helpful to facilitate implementation, the third outcome measure was the consensus between the first two outcome measures (relevance and year of entrustment of each EPA) assessed in the preced-ing (institutional validation) and current study.”

Statistical analysis is mentioned and the Presentation of results in table 1 is clear (Table 1. Content Validity Indices (CVI) and the year (YR) of entrustment of the EPAs), however, the basic demographic data of partici-pants in the study (age, gender, education, background, work experience level, etc) is not mentioned. It’s im-portant to mention the first table before the study. Sometimes, this data is linked to the outcome of study measures.

Response: Thank you very much for this relevant and helpful comment. As stated before, we have included a new table (table 1, page 8), in which we mention the basic demographic data (as far as assessed and available) from all study participants of both studies. We believe this table has an added value for the manuscript and hope to also meet your satisfaction. Please find table 1 on page 8.

Data satisfies the objectives of the study, all subjects/Variables in the study are accounted for appropriate statistical tests are applied, and Data Analysis is correctly interpreted. The alignment process led to four major changes in the curriculum is explicit. However, Figures 1 and 2 are not illegible in this pdf so cannot comment. Make it in readable format.

Response: Thank you very much. We uploaded the Figures as TIF. files- because it is required on the submitting guidelines. We have additionally uploaded the figures in editable formats. Please let us know if any other for-mat is needed.

Limitations of the study mentioned, however, there are no comments on the generalization of the results in medical education.

Response: Thank you very much for this comment. Indeed, the content of the curriculum is not generalizable to the broad field of medical education. Alas, we don’t see the content of the curriculum as the main value of our work- the true value of our work lies within how we set the basis for unification of training (anesthesiology) and for the transformation of time-based into EPA-based training. Our approach is of great significance and hence can be transferred to the field of medical education in general. We have highlighted this aspect in the discus-sion. Please find the paragraph on page 19 lines 451-454:

“Although the content of our curriculum itself is not generalizable to medical education, the value of the approach of our study still merits consideration. This approach could assist other medical disciplines- in graduate and undergraduate medical education- with the integration of EPAs into training.” 

Written in the required style however, Mostly are more than 5 years old so use the latest research studies, especially in the discussion part, and add Local literature.

Response: Thank you very much for this helpful comment- which is directly linked to a challenge. Actually, there are very few current studies related to EPAs. This is also an indirect benchmark for how urgent new studies on this field are and adds to the value of our work. If we do not continue research on this field, the concept of EPA will remain a theoretical construct. We have added the latest research and position papers in the introduction and discussion section and also included local literature. The latest research states that the need for EPA curric-ula is present (Burkhardt et al., Flentje et al.), that some are being developed (Dabbagh et al.) and that the concept of EPA is often not known well and therefore not applied in the clinical workplace (Ruiz et al.). We have embedded the current literature within our manuscript adequately and hope that we meet your satisfaction and that we could highlight again the novelty and need of our work.

Few Grammatical mistakes need careful review.

Response: Thank you. We have double checked the manuscript with a native speaker and believe that we have identified and corrected the mistakes.

Thank you very much for all of your comments. If there is anything more you wish to be changed we will be happy to provide further amendments.

---

## [Decision Letter · Decision Letter 1]

21 Jun 2023

National consensus on Entrustable Professional Activities for competency-based training in Anaesthesiology

PONE-D-22-30955R1

Dear Dr. Moll-Khosrawi,

We’re pleased to inform you that your manuscript has been judged scientifically suitable for publication and will be formally accepted for publication once it meets all outstanding technical requirements.

Kind regards,

Valérie Pittet, PhD

Academic Editor

PLOS ONE

Additional Editor Comments (optional):

The authors have appropriately answered all comments. The paper has been greatly improved.

Reviewers' comments:

Reviewer's Responses to Questions

**Comments to the Author**

1. If the authors have adequately addressed your comments raised in a previous round of review and you feel that this manuscript is now acceptable for publication, you may indicate that here to bypass the “Comments to the Author” section, enter your conflict of interest statement in the “Confidential to Editor” section, and submit your "Accept" recommendation.

Reviewer #1: All comments have been addressed

2. Is the manuscript technically sound, and do the data support the conclusions?

Reviewer #1: Yes

3. Has the statistical analysis been performed appropriately and rigorously? 

Reviewer #1: Yes

4. Have the authors made all data underlying the findings in their manuscript fully available?

Reviewer #1: Yes

5. Is the manuscript presented in an intelligible fashion and written in standard English?

Reviewer #1: Yes

6. Review Comments to the Author

Reviewer #1: This is a unique study that the authors did great effort to show their results. The type of study is not discussed before.

7. PLOS authors have the option to publish the peer review history of their article (what does this mean?). If published, this will include your full peer review and any attached files.

Reviewer #1: No

---

## [Editor Report · Acceptance letter]

3 Jul 2023

PONE-D-22-30955R1 

National consensus on Entrustable Professional Activities for competency-based training in Anaesthesiology 

Dear Dr. Moll-Khosrawi:

I'm pleased to inform you that your manuscript has been deemed suitable for publication in PLOS ONE. Congratulations! Your manuscript is now with our production department. 

Kind regards, 

on behalf of

PD Dr. Valérie Pittet 

Academic Editor

PLOS ONE